# REVISIT LARGE-SCALE IMAGE-CAPTION DATA IN PRE-TRAINING MULTIMODAL FOUNDATION MODELS

**Zhengfeng Lai,** * **Vasileios Saveris,** * **Chen Chen, Hong-You Chen, Haotian Zhang**
**Bowen Zhang, Juan Lao Tebar, Wenze Hu, Zhe Gan, Peter Grasch**
**Meng Cao, Yinfei Yang** [†]
Apple
{jeff_lai,v_saveris,pgrasch,mengcao,yinfeiy}@apple.com

## ABSTRACT

Recent advancements in multimodal models highlight the value of rewritten captions for improving performance, yet key challenges remain. For example, while synthetic captions often provide superior quality and image-text alignment, it is not clear whether they can fully replace AltTexts: the role of synthetic captions and their interaction with original web-crawled AltTexts in pre-training is still not well understood. Moreover, different multimodal foundation models may have unique preferences for specific caption formats, but efforts to identify the optimal captions for each model remain limited. In this work, we propose a novel, controllable, and scalable captioning pipeline designed to generate diverse caption formats tailored to various multimodal models. By examining Short Synthetic Captions (SSC) towards Dense Synthetic Captions (DSC+) as case studies, we systematically explore their effects and interactions with AltTexts across models such as CLIP, multimodal LLMs, and diffusion models. Our findings reveal that a hybrid approach that keeps both synthetic captions and AltTexts can outperform the use of synthetic captions alone, improving both alignment and performance, with each model demonstrating preferences for particular caption formats. This comprehensive analysis provides valuable insights into optimizing captioning strategies, thereby advancing the pre-training of multimodal foundation models.

## 1 INTRODUCTION

Large-scale image-text datasets have been crucial in advancing multimodal foundation models. For instance, CLIP (Radford et al., 2021) is pre-trained on 400 million image-text pairs collected from the Web. However, web-crawled data, particularly AltText, often suffer from insufficient visual details and noisy content, as illustrated in Fig. 1. Recent studies highlight the benefits of synthetic captions, which provide better image-text alignment and improved data quality. Research on LaCLIP (Fan et al., 2024) and ShareGPT4V (Chen et al., 2024a) demonstrates that synthetic captions can improve the performance of CLIP and multimodal large language models (MLLMs), respectively. This raises a key question: if higher-quality synthetic captions can be generated, could they fully replace web-crawled AltText? Should we consider disregarding AltText altogether?

To investigate this question, we first adopt the approach of VeCLIP (Lai et al., 2024) and train CLIP using synthetic captions generated by LLaVA (Liu et al., 2023b). Similar to the results discussed in Li et al. (2024b), training CLIP fully on synthetic captions of higher quality degrades CLIP's performance significantly: as shown in Fig. 2, when compared to using only AltText, the use of LLaVA captions results in a substantial drop on zero-shot ImageNet classification tasks. However, after combining original noisy AltText and LLaVA captions, we achieve the best results in both classification and retrieval tasks. This observation raises a critical question: what constitutes the optimal image-text data for multimodal foundation models? Despite its importance, research on the interplay between synthetic captions and AltText remains limited. Our findings suggest that while rewriting techniques can enhance image-text alignment, they may reduce data diversity due to

---

*Equal contribution.
[†]Corresponding author.

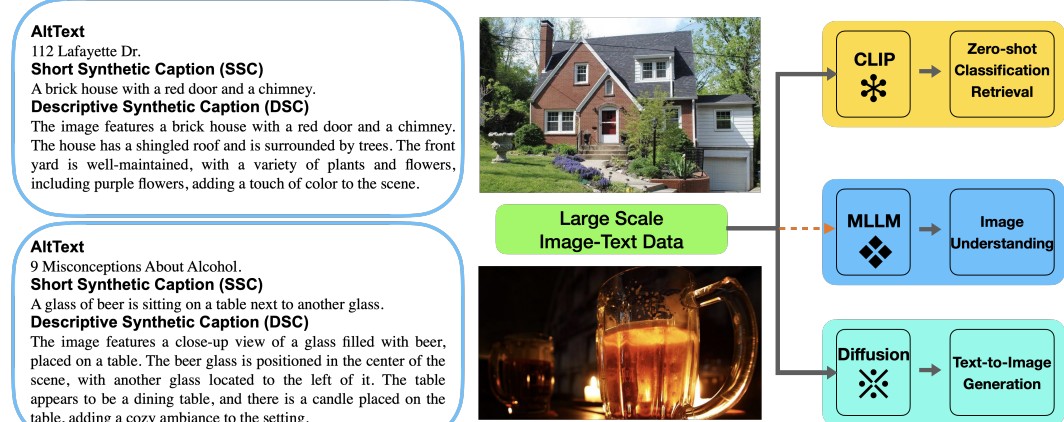

Figure 1: The role of image-text data in multimodal foundation models: a **key** component in training CLIP and Diffusion Model, and **essential** for multimodal LLM (MLLM) pre-training alongside text and interleaved image-text data. We propose a controllable captioning pipeline to synthesize different types of captions and explore optimal image-text data recipes for training these foundation models.

dependence on a limited set of LLMs or MLLMs for caption generation. Specifically, since CLIP is a foundational vision model that benefits from learning diverse concepts, relying on synthetic captions can potentially hinder CLIP's training due to lack of diversity in vocabulary and mentioned concepts (Fan et al., 2024).

In addition to the role of AltText, another open question concerns the optimal formats for synthetic captions. For instance, advanced multimodal models like LLaVA-NeXT (Li et al., 2024a) indicate that recaptioned datasets are advantageous during stages focused on high-quality knowledge acquisition. DALL-E 3 (Betker et al., 2023) demonstrates that using 95% synthetic captions can yield superior results, particularly when the captions are highly descriptive. Similarly, MM1 (McKinzie et al., 2024) shows that even a small fraction (7%) of high-quality caption data can significantly boost few-shot performance. Given these insights, our work focuses on two key unresolved questions: *1) What is the role and value of synthetic captions, and how do they interact with the original AltText? 2) What types of synthetic captions are most effective for different foundation models?* To address the first question, we revisit why prior works (Betker et al., 2023; Lai et al., 2024; McKinzie et al., 2024) continue using noisy web-crawled AltText, even when rewritten captions are available during training. Intuitively, since CLIP is a straightforward model pre-trained on image-text pairs, highly aligned captions should be advantageous. However, relying solely on synthetic captions may actually degrade CLIP's performance, as shown in Fig. 2(a). For the second question, we investigate the effects of Short Synthetic Captions (SSC) and Descriptive Synthetic Captions (DSC) on CLIP. As depicted in Fig. 2(b), surprisingly, more descriptive captions yield inferior results compared to shorter captions for CLIP training, despite their greater detail.

To explore these insights further to address the two questions, we introduce a novel, controllable, and scalable captioning pipeline that enables the generation of diverse caption formats at scale, tailored to the specific needs of different multimodal foundation models. Our pipeline is designed to build large-scale image-text data for the pre-training stage in a scalable way. With this pipeline, we use SSC and DSC as two main examples on how to customize the captioning format. Our pipeline can serve as a cost-effective alternative to GPT-4V for generating high-quality captions. To solve the second question, we conduct a comprehensive study on the effectiveness of different types of synthetic captions across a range of foundational models and downstream tasks. Our approach involves a systematic evaluation of various captioning strategies, including SSC, DSC, and mixed training methods that combine original AltText with synthetic data. We seek to determine the optimal captioning techniques for specific models, such as CLIP, diffusion models, and multimodal LLMs, and to assess their impact on both model performance and data diversity. Furthermore, we investigate the interaction between synthetic captions and original AltText, analyzing whether a hybrid approach can balance the need for diverse data with the benefits of enhanced image-text alignment.

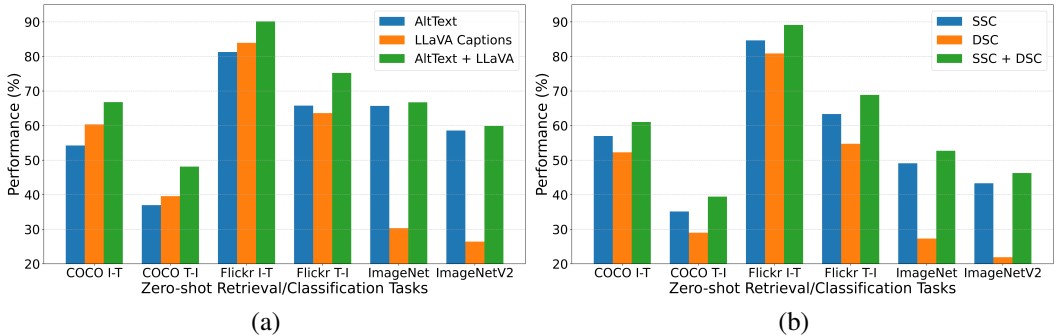

Figure 2: Zero-shot retrieval and classification performance of CLIP models. (a) The effect of synthetic captions (LLaVA recaptioned) and AltText: solely using LLaVA captions can improve retrieval tasks but significantly deteriorate the zero-shot classification performance. (b) The effect of different formats of synthetic captions on CLIP: Short Synthetic Captions (SSC) show superior results to Descriptive Synthetic Captions (DSC) and the combination of them achieves the best results.

Overall, our contributions are summarized as follows.

- We explore the MLLM as the image describer and present a controllable and human-aligned captioning pipeline to convert MLLM into an image captioner.

- We synthesize several formats of captions including Short Synthetic Captions (SSC) towards Dense Synthetic Captions (DSC+), then conduct extensive pre-training experiments to systematically study the role of synthetic captions and their intersection with original AltText across three multimodal foundation models.

- We verify the image-caption training recipe that 1) AltText provide data variety and synthetic captions provide better image-text alignment, 2) different foundation models have their own preferred formats, which highlights the necessity and importance of the controllable captioning pipeline in building multimodal foundation models.

## 2 RELATED WORK

**Multimodal Foundation Models.** CLIP (Radford et al., 2021) is one of the pioneering multimodal foundation models connecting image and text. By training on 400 million image-text pairs, CLIP shows strong zero-shot capabilities. It lays the groundwork for the development of more advanced multimodal foundation models, such as multimodal large language models (MLLMs) (Liu et al., 2023b; Wang et al., 2023; Chen et al., 2024b; Tong et al., 2024) for vision-language understanding and diffusion models (Rombach et al., 2022; Podell et al., 2023) for text-to-image generation. These advanced models often utilize CLIP's vision tower as their vision encoder.

**Improving Image-Text Data.** Web-crawled image-text data often suffer from issues like image-text misalignment and poor-quality textual descriptions (Lai et al., 2024; Li et al., 2024b). There are two common ways for improving image-text data: 1) data filtering-based methods remove low-quality data such as misaligned image-text pairs by human-assisted systems (Yu et al., 2024a; Sun et al., 2023) or pre-trained models (Li et al., 2022b; Schuhmann et al., 2021; Gadre et al., 2024; Fang et al., 2023); 2) data recaptioning-based methods usually leverage a LLM rewrite the original caption or a MLLM to recaption a caption on the image. For example, ShareGPT4V (Chen et al., 2024a) uses GPT-4V to write highly descriptive captions for their images. LaCLIP (Fan et al., 2024) leverages several LLMs to rewrite captions with different writing styles for data diversity. Recap-DataComp-1B (Li et al., 2024b) uses a LLaMA-3 based model to scale the captions. Different from the aforementioned works, we mainly focus on generating different types of captions and exploring 1) the format of ideal captions needed for each multimodal foundation model and 2) a systematic analysis of the intersection between AltTexts and synthetic captions.

**Short Synthetic Caption (SSC):**

A pronghorn antelope standing in a field of tall grass.

**Dense Synthetic Caption (DSC+):**

A medium close-up front view of a pronghorn antelope standing on a ground floor made up of tall green grass. The antelope is facing forward, it has a light brown body, a white underbelly, and black horns that curve up and stick out of its head. The antelope is surrounded by a blurry view of an open desert environment that consists of green plants, small bushes, and tall grass. Towards the top right, a blurry view of a large rock hill can be seen.

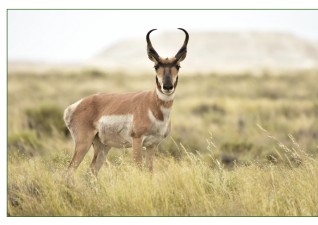

**AltText Fusion Caption (AFC):**

A photograph of a pronghorn standing in a field of tall grass. The pronghorn is looking straight at the camera. The background is a hazy sky and a distant hill. This pronghorn is on Seedskadee National Wildlife Refuge in Wyoming, USA, managed by USFWS Mountain Prairie.

**Descriptive Synthetic Caption (DSC):**

A photograph of a pronghorn antelope standing in a field of tall grass. The antelope is facing the camera and its horns are prominently featured. The background consists of a hazy sky and a field of tall grass.

***AltText:*** *Pronghorn on Seedskadee National Wildlife Refuge by USFWS Mountain Prairie.*

Figure 3: An example of controllable captions of diverse formats generated by our captioner: we can generate from brief to dense description and fuse AltText into the caption (AFC).

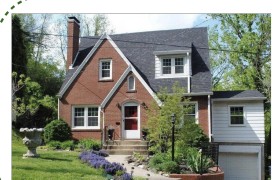

**LLaVA-v1.5-13B**
**Q: Describe the image with three sentences.**

A: The image features a red brick house with a black roof, surrounded by a lush green yard. The house has a chimney and a red door, adding to its charm. The yard is filled with various plants, including a potted tree. The scene also includes a car parked in front of the house, and a fire hydrant nearby.

**ShareGPT4V-7B**
**Q: Describe the image with three sentences.**

A: The image features a large brick house with a black roof, situated on a hill. The yard is filled with various plants and flowers. In addition to the house, there is a garage located on the right side of the property. The scene also includes a bird, possibly a pigeon, perched on the roof of the house, adding a touch of nature to the scene.

Figure 4: Directly using MLLMs as image captioners may result in hallucinations and generate captions that do not align with specific instructions: both LLaVA (Liu et al., 2023b) and ShareGPT4V (Chen et al., 2024a) generates over three sentences and obvious hallucination.

## 3    CUSTOMIZED RE-CAPTIONING FOR MULTIMODAL FOUNDATION MODELS

Image-text data are fundamental to the success of multimodal foundation models, serving as a bridge between visual and textual representations. For example, CLIP (Radford et al., 2021) is pre-trained on 400M web-crawled image-text pairs, enabling it to learn rich, transferable representations that can be applied to various downstream tasks. The importance of precise and detailed captioning is further highlighted in LLaVA-Next (Li et al., 2024a), where re-captioned detailed descriptions are utilized in training Stage-1.5, enhancing the model's ability to understand and generate nuanced content. Similarly, DALL-E 3 (Betker et al., 2023) shows that the prompt-following capabilities of text-to-image models can be significantly improved by training on highly-descriptive generated image captions. This shows the critical role of captions in shaping a model's capacity to align visual and textual information, ultimately improving performance across a wide range of multimodal tasks. However, the optimal captioning strategy for different foundational models remains under-explored. To address this gap, we introduce a novel, controllable, and scalable captioning pipeline designed to generate diverse caption formats at scale, supported by evaluation metrics that ensure high CLIP scores and minimal hallucination. We summarize the capability of our captioning model by generating the following formats of captions as shown in Fig. 3:

**Short Synthetic Caption (SSC)**: a concise sentence that describes the primary subject of the image.

**Descriptive Synthetic Caption (DSC)**: a description limited to 78 tokens, emphasizing the central subject and key visual elements.

**Dense Synthetic Caption (DSC+)**: a more comprehensive description detailing the main subject along with the background, setting, and any significant objects or actions.

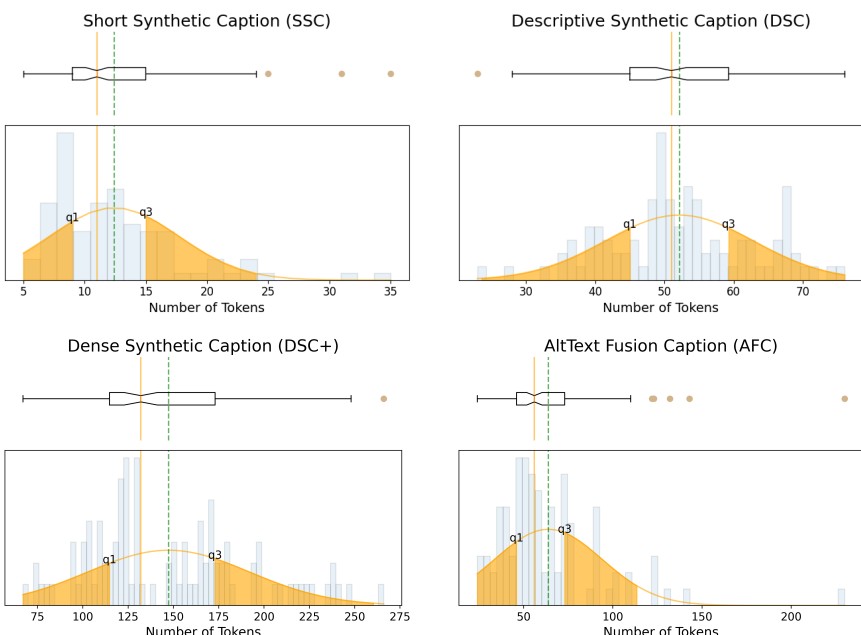

Figure 5: Distribution of token lengths of our generated captions in four formats: we caption COCO-2017 images and visualize their distributions.

**AltText Fusion Caption (AFC)**: a caption similar to DSC, but integrated with AltText where appropriate. This type of caption removes unnecessary details often found in AltText, offering a cleaner, more cohesive description than a simple concatenation of AltText and synthetic caption.

### 3.1 MLLM AS AN IMAGE DESCRIBER

VeCLIP (Lai et al., 2024) employs LLaVA (Liu et al., 2023b) for image captioning, while ShareGPT4V (Chen et al., 2024a) utilizes GPT-4V for this task. Compared to traditional image captioners like BLIP (Li et al., 2022a), MLLMs offer several advantages and are natually good image describers. MLLMs can generate longer and more detailed captions, as demonstrated by LLaVA-Next (Li et al., 2024a), where a 34B model was used to produce highly descriptive captions. However, directly using MLLM as an image describer may have two major limitations: 1) MLLM may not strictly follow the instruction to generate a specific format of caption (Liu et al., 2023a); 2) these instruction fine-tuned MLLM tends to have hallucination. As shown in Fig. 4, both LLaVA (Liu et al., 2023b) and ShareGPT4V (Chen et al., 2024a) fails to describe the image using only three sentences but generate hallucinated contents (highlighted in red). Although GPT-4V shows stronger capability and many works use it for captioning, the scalability remains limited due to the cost. Therefore, in this work, we focus on building a cost-effective captioner instead of using GPT-4V.

To alleviate the above two limitations, we first investigate the origins of hallucination in MLLMs, hypothesizing two primary sources: 1) inherent limitations of the LLM and 2) the quality of supervised fine-tuning (SFT) datasets, which often are itself synthetically derived or processed. We focus on the latter, proposing that mitigating hallucinations at the dataset level is essential for converting an MLLM into an effective captioner. We also address the format-following issue by fine-tuning the MLLM on a curated captioning-specific dataset, transforming it into a purpose-built captioning model to generate diverse captions.

### 3.2 TWO-STAGE HUMAN-ALIGNED CAPTIONING

**Stage 1: transforming MLLM into a customized captioner.** To minimize hallucinations from MLLMs, we begin by constructing a clean and precise fine-tuning dataset. Instead of relying on GPT-4 generated data, we curate a high-quality dataset of human-annotated image-text pairs, named Stage-1-1M. This dataset contains short, human-curated captions, with five captions per image. Additionally, we integrate an OCR detection model to extract in-image text, which, alongside the

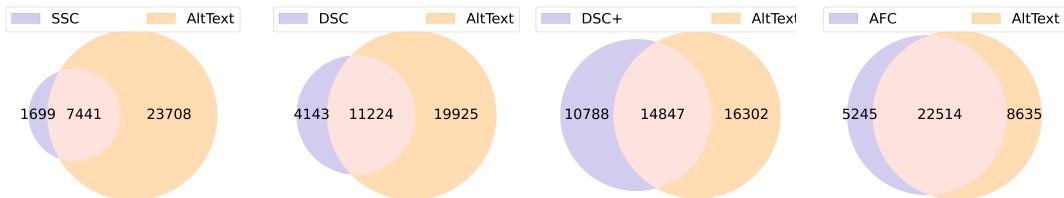

Figure 6: The number of unique entities in different synthetic captions (randomly sample 17.5k images) compared to AltText: AltText provides more unique entities as wider knowledge.

captions, is fed into a large language model (LLM) for summarization and controlled rewriting. By employing a strict prompt, we prevent the LLM from introducing extraneous information, while few-shot prompts guide the model in generating various caption formats, including both concise and descriptive styles. We further enhance the dataset through post-processing, using heuristic and model-based quality checks to improve its overall quality. This comprehensive fine-tuning on the Stage-1-1M dataset transforms the MLLM, specifically the 3B version of MM1 (McKinzie et al., 2024), into a customized captioner that aligns closely with the intended output characteristics.

**Stage 2: Human-aligned further fine-tuning.** While the Stage-1-1M dataset effectively establishes a foundation, it lacks the depth required for more descriptive captioning tasks. In Stage 2, we address this by incorporating descriptive human-annotated data to enhance caption diversity and quality. We curate a new dataset, named Stage-2-HA, specifically designed for detailed captioning. This dataset is meticulously annotated by human experts to capture nuanced visual elements and complex scene descriptions. Post-annotation, we leverage an LLM to reformat these captions into multiple stylistic variations, including both concise and richly descriptive formats. By imposing strict constraints during the LLM processing, we ensure alignment with human-generated content, avoiding the pitfalls of hallucination. The captioner is then fine-tuned on the Stage-2-HA dataset, resulting in a highly refined and human-aligned captioning model capable of generating captions tailored to specific use cases. This dual-stage fine-tuning process not only enhances the model's adaptability but also ensures a balance between brevity and descriptiveness, making it a versatile tool in controllable captioning. The overview figure can be found in Appendix.

### 3.3 CAPTION ANALYSIS

**Richness assessment: token length and Average Number of Assertions.** Fig. 5 illustrates the distribution of the number of tokens of various caption types generated in this study. Specifically, SSC mainly ranges from 10 to 15 tokens, while DSC spans from 40 to 60 tokens, both fitting within the text encoder's capacity in CLIP. In contrast, most DSC+ captions exceed 100 tokens. Besides that, we propose Average Number of Assertions (ANA) to quantify the richness of captions. We prompt an LLM to generate different assertions of a caption to analyze our different formats of captions in terms of the richness. More details of this approach is in Appendix. Note that the ANA for SSC is 2.49, DSC as 8.13 and DSC+ as 12.20, showing more visual contents.

**Diversity assessment: number of unique entities in captions.** We hypothesize that the original, albeit noisy, AltText may carry a broader range of diverse information and knowledge, offering potential advantages for CLIP's pre-training. To assess this diversity, we quantify the number of unique entities present in the captions. As shown in Fig. 6, AltText contains a higher number of unique entities, which could be beneficial in providing a wider knowledge base. Compared to SSC, more descriptive captions also contain more entities.

## 4 IMAGE-CAPTION DATA FOR MULTIMODAL FOUNDATION MODELS

We mainly discuss three foundation models: CLIP, multimodal LLM, and diffusion models. For both CLIP and diffusion models, since the text encoder is limited to 77 tokens (Zhang et al., 2024), we focus primarily on SSC and DSC. For the multimodal LLM, we explore more detailed versions, including DSC+ and AFC. We summarize our key findings below:

Table 1: Effect of different synthetic captions on CLIP with ViT-B/16 as the backbone.

| Pre-train Caption | COCO (R@1) I-T | COCO (R@1) T-I | Flickr30k (R@1) I-T | Flickr30k (R@1) T-I | ImageNet | ImageNetV2 |
|---|---|---|---|---|---|---|
| AltText | 54.24 | 36.98 | 81.30 | 65.80 | 65.70 | 58.58 |
| DSC | 52.28 | 29.00 | 80.90 | 54.75 | 27.30 | 21.91 |
| SSC | 57.00 | 35.15 | 84.67 | 63.35 | 49.10 | 43.31 |
| AFC | 54.82 | 34.84 | 84.00 | 62.18 | 38.98 | 35.11 |
| DSC + AltText | 65.84 | 46.08 | 90.26 | 73.94 | 66.18 | 58.74 |
| SSC + AltText | 66.67 | 48.13 | 91.81 | 76.54 | **66.63** | **59.57** |
| AFC + AltText | 63.98 | 43.76 | 89.10 | 73.32 | 66.47 | 58.84 |
| All Synthetic + AltText | **70.12** | **50.21** | **93.00** | **77.72** | 64.91 | 57.92 |

Table 2: Evaluation with linear probing on ImageNet for CLIP.

| Pre-train Caption | Zero-shot | Linear Probing | Gain |
|---|---|---|---|
| AltText | 65.70 | 78.34 | +12.64 |
| DSC | 27.30 | 75.72 | +48.42 |
| SSC | 66.63 | 79.94 | +13.31 |
| AFC | 38.98 | 75.53 | +36.55 |
| DSC + AltText | 66.18 | 79.96 | +13.78 |
| SSC + AltText | 66.63 | 79.94 | +13.31 |
| AFC + AltText | 66.47 | 78.70 | +12.23 |
| DSC + SSC + AltText | 65.16 | 80.01 | +14.85 |
| All Synthetic + AltText | 64.91 | 79.55 | +14.64 |

- The tradeoff between the richness of captions and their accuracy needs to be balanced based on the multimodal tasks.

- Both AltText and synthetic captions are important for CLIP's training, with shorter captions yielding better performance. Linear probing is an additional effective way to evaluate the representations.

- Pre-training and SFT benchmarks can behave differently in multimodal LLMs. On the SFT benchmark, MM1 shows a preference for DSC+ alone.

- In the diffusion model, DSC emerges as the most effective captioning strategy.

## 4.1 IMAGE-CAPTION DATA FOR CLIP

We use VeCap-300M (Lai et al., 2024), a web-crawled dataset with raw AltText as our main pre-training dataset for CLIP. Besides AltText, we generate several synthetic captions for the study. Then, we use ViT-B/16 as the vision encoder. The training details can be found in Appendix.

**Effect of synthetic captions.** We first study the effect of Short Synthetic Captions (SSC) and Descriptive Synthetic Captions (DSC). We summarize the results in Table 1. Interestingly, while SSC enhances retrieval performance, it leads to a substantial drop in zero-shot ImageNet accuracy. Moreover, despite demonstrating that synthetic captions have superior image-text alignment and reduced noise, the more descriptive captions (DSC) perform worse across all benchmarks. Based on these results, the descriptive version appears suboptimal for CLIP training, leading to inferior performance. For example, despite DSC capturing more visual concepts within the captions, its zero-shot performance shows a significant degradation of 21.8% compared to SSC. We hypothesize this performance drop may be partially due to a distribution mismatch as the prompts in COCO/Flickr/ImageNet are short (e.g., "a photo of {}"). To further explore the performance gap between DSC and SSC on CLIP, we also use linear probing, which provides a direct measure of the quality and generalization capability of the representations learned by CLIP. Strong performance from a linear classifier on specific tasks indicates that the pre-trained model has effectively captured relevant and discriminative features, underscoring the robustness of its embeddings. We summarize the results on linear probing in Fig. 2: even though DSC and SSC shows lower zero-shot performance, they achieve comparable results to AltText after linear probing, indicating similar pre-trained representations. This indicates that the relatively poor zero-shot results of DSC can be significantly improved with linear probing, implying that DSC's representations are richer than initially presumed.

**Intersection of synthetic captions and AltText.** Table 1 shows training CLIP solely on large-scale synthetic captions may get inferior results compared to the original AltText, even though synthetic captions have better image-text alignment. We hypothesize that synthetic captions rewritten from a MLLM may hurt the diversity and knowledge coverage of the original AltText. Therefore, we blend our synthetic captions and AltText, such as DSC + AltText and SSC + AltText: there is a significant boost in retrieval performance—e.g., over a 10% improvement on COCO. Additionally, SSC + AltText also leads to gains in ImageNet accuracy. Utilizing a mixture of all synthetic caption formats yields the best performance in retrieval tasks, as visualized in Fig. 6.

**Optimal mixture ratio between synthetic captions and AltText.** Considering the wider knowledge of AltText (Fig. 6) and the better alignment of synthetic captions, we explore the optimal mixture ratio. Specficially, we use SSC and AltText from VeCap-300M (Lai et al., 2024) as an example, with results shown in Fig 7. A ratio of 0 corresponds to using only SSC, while 100 corresponds to using only AltText. We observe that CLIP achieves optimal performance across both retrieval and classification tasks when the ratio is tuned to around 40-50%. A lower proportion of AltText leads to

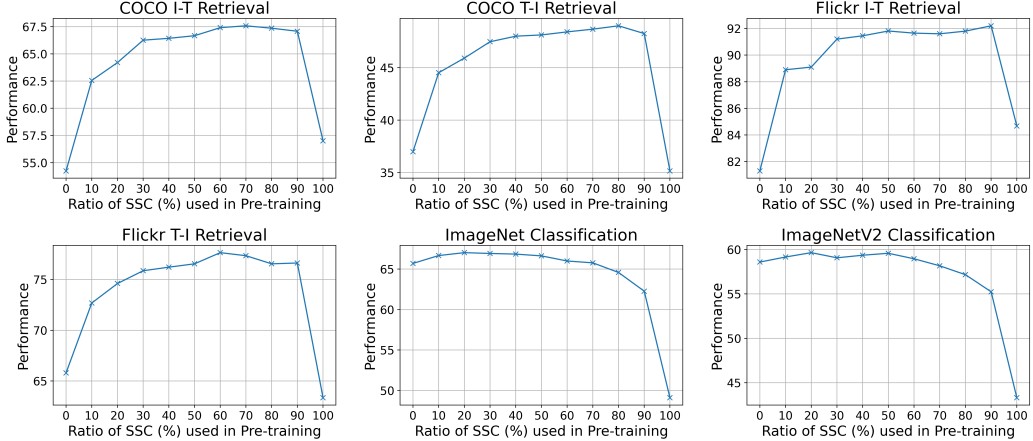

Figure 7: The intersection of synthetic captions and AltText for CLIP: we increase the proportion of SSC mixed with AltText during training. All experiments use ViT-B/16 as the backbone and VeCap-300M (Lai et al., 2024) as the pre-training dataset.

a drop in retrieval performance, whereas a lower proportion of SSC results in decreased accuracy in ImageNet zero-shot classification. This observation aligns with findings in (Li et al., 2024b). From this, we verify that AltText provides broader knowledge coverage and greater diversity, which benefits CLIP's pre-training by enabling the model to grasp a wider range of concepts. This diversity may serve as a foundation for generalization, allowing CLIP to better represent varied contexts and domains in zero-shot classification.

**Exploration of the optimal way of using AltText: AFC vs simple mixture.** As shown in Fig. 7 and Table 1, we verify that a simple mixture of our synthetic captions and AltText can achieve superior results. In addition to this straightforward combination, we investigate alternative methods of incorporating AltText into the training process. One promising approach is to fuse the knowledge from AltText directly into the synthetic captions. To enable this, we fine-tune our captioner with instruction-following capabilities, generating AltText Fusion Captions (AFC). While AFC shows improvement over DSC due to the enriched AltText information, it underperforms when compared to the simple mixture of DSC and AltText.

## 4.2 IMAGE-CAPTION DATA FOR MULTIMODAL LLM

Table 3: The effect of synthetic captions on MM1 (1.2B model) pre-training.

| Pre-train Caption | TextCore | 0-Shot | 4-Shot | 8-Shot |
|---|---|---|---|---|
| AltText | 53.48 | 34.80 | 55.81 | 59.72 |
| DSC | **53.76** | 37.05 | 59.36 | 63.68 |
| DSC + AltText | 53.71 | 37.09 | **60.31** | **63.96** |
| SSC | 53.46 | **37.35** | 59.19 | 63.56 |
| SSC + AltText | 53.20 | 37.16 | 58.22 | 62.22 |

Table 4: The ratio ablation on MM1 pre-training between synthetic captions and AltText.

| Mixing Ratio | TextCore | 0-Shot | 4-Shot | 8-Shot |
|---|---|---|---|---|
| 33/66 | 53.86 | 35.16 | 59.69 | 63.97 |
| 50/50 | 53.86 | 36.40 | 60.17 | 64.13 |
| 66/33 | **54.24** | **37.02** | 60.26 | **64.71** |
| 80/20 | 53.96 | 35.74 | 60.09 | 64.69 |
| 100/0 | 54.19 | 15.54 | 60.24 | 64.02 |

As we study the large-scale image-caption data, we use MM1 (McKinzie et al., 2024) as one example and focus on the pre-training stage. MM1 (McKinzie et al., 2024) claims that captioning data lift the zero-shot performance and synthetic captions are helpful for few-shot learning. Based on this insight, we further study the captioning data recipe between original AltText and synthetic captions. We follow the pre-training setup and the evaluation benchmark in MM1 (McKinzie et al., 2024) to report TextCore and 0/4/8-shot performance. All of the experiments are conducted on the 1.2B model. We pre-train the model with 50K steps and the batch size is 512. More details are in Appendix.

**Effect of synthetic captions for pre-training benchmark.** We generate DSC and SSC captions for VeCap-300M (Lai et al., 2024), as used in MM1 (McKinzie et al., 2024), and replace the captions during MM1's pre-training. The results, summarized in Table 3, show that our synthetic captions yield improved performance in vision-text benchmarks across 0-shot to 8-shot settings. For example,

Table 5: SFT evaluation of models pre-trained with different types of captions. We use the same SFT evaluation benchmarks as in MM1 (McKinzie et al., 2024). (* Concatenation of two captions.)

| Pre-Trained Data | VQAv2 | VQAT | MMMU | MathV | MMEP | MMEC | SEED | POPE | LLaVAW | Average |
|---|---|---|---|---|---|---|---|---|---|---|
| AltText | 77.1 | 66.1 | 31.4 | 28.3 | 781.7 | 225.4 | 61.7 | 84.6 | 69.8 | 57.6 |
| LLaVA Caption | 78.0 | 65.4 | 30.0 | 27.6 | 773.4 | 200.4 | 63.5 | 83.7 | 63.6 | 56.5 |
| SSC | 78.8 | 65.8 | **33.6** | 28.2 | 760.2 | 216.4 | 63.1 | 84.4 | 69.0 | 57.9 |
| DSC | 77.1 | 61.8 | 30.8 | 27.9 | 596.6 | 213.6 | 64.6 | 83.9 | 70.8 | 56.3 |
| DSC+ | 79.0 | 65.4 | 32.6 | 29.4 | 727.2 | 224.3 | **66.8** | 85.1 | 71.5 | **58.7** |
| AltText + SSC (*) | 77.7 | 65.4 | 32.3 | 27.6 | **781.9** | **236.1** | 62.0 | 84.2 | 70.3 | 57.7 |
| AltText + DSC (*) | 78.2 | 66.9 | 31.9 | **30.7** | 686.4 | 216.4 | 63.8 | 84.3 | **72.0** | 58.2 |
| AltText + DSC+ (*) | 79.2 | **66.9** | 31.1 | 29.6 | 740.2 | 229.3 | 65.5 | 85.1 | 68.1 | 58.2 |
| AFC | 78.0 | 65.6 | 32.3 | 29.4 | 713.2 | 214.3 | 66.0 | 84.4 | 70.0 | 58.0 |
| SSC + DSC + DSC+ | **79.4** | 66.5 | 30.2 | 30.3 | 689.3 | 198.9 | 65.5 | 83.8 | 67.5 | 57.5 |
| AFC + SSC + DSC + DSC+ | 77.3 | 62.6 | 31.6 | 27.7 | 661.5 | 198.2 | 64.1 | 83.5 | 64.2 | 55.9 |

SSC achieves a **+1.3%** performance gain in 0-shot evaluation compared to the original MM1. Unlike CLIP's experiments, the descriptive captions (DSC) outperform SSC, with the combination of DSC and original AltText delivering the best results in this context. We also conduct a ratio ablation study on MM1 pre-training to explore the optimal balance between synthetic captions and AltText, as summarized in Table 4. The results indicate that a 66/33 mixing ratio yields the best overall performance across all evaluation settings. Specifically, this ratio achieves the highest scores for TextCore (54.24), 0-shot (37.02), 4-shot (60.26), and 8-shot (64.71) tasks. While increasing the proportion of synthetic captions generally improves performance, there is a significant drop in 0-shot performance when using only synthetic captions (100/0 ratio).

**Effect of synthetic captions for SFT benchmark.** Besides pre-training benchmark, we also conduct SFT and then evaluate the model to analyze the profound effect of image-caption data in multimodal LLMs. We fix the SFT recipe to have a fair comparison. As shown in Table 5, DSC+ and the concatenation of AltText with DSC+ deliver the best performance. This strongly suggests the importance of detailed captions, despite them containing potentially the highest number of hallucinations among all the caption types we test. On the other hand, concatenating AltText with synthetic captions does not yield significant improvement in the SFT benchmark, contrasting with the gains observed in pre-training benchmarks. We suggest that the primary role of image-caption data during the pre-training phase of multimodal LLMs is to enhance image-text alignment. Consequently, more detailed captions, such as DSC+, deliver superior results after the SFT stage.

**DSC+ alone can outperform diverse synthetic captions.** Unlike CLIP, where mixing diverse synthetic captions leads to superior results, for MM1, detailed captions (DSC+) alone yield the best performance after the SFT stage. As shown in Table 5, DSC+ achieves a 58.7% score, outperforming LLaVA captions (56.5%) by 2.2%. This suggests that providing richer and more specific information in captions helps multimodal LLMs like MM1 generalize better after the SFT stage. The combination of SSD, DSC, DSC+, and AFC does not lead to better results, suggesting that multimodal LLMs may benefit more from detailed captions during pre-training. These findings suggest that while combining synthetic captions proves beneficial in some contexts (e.g., CLIP), for multimodal LLMs like MM1, a single, detailed caption offers more effective guidance during pre-training.

## 4.3 IMAGE-CAPTION DATA FOR DIFFUSION MODEL

Inspired by DALLE-3 (Betker et al., 2023), detailed and short captions can improve the prompt following ability. In this work, our DSC not only covers the main objects within the scene, but also their relationships, attributes, and the broader context in which they are situated. We hypothesize that this level of detail allows the model to generate images that are not only visually accurate but also semantically aligned with the textual input. We implement Stable Diffusion 3 (Esser et al., 2024) and use this diffusion model as our studying example on text-to-image generation tasks. The backbone is based on the DiT architecture (Peebles & Xie, 2023) that focuses exclusively on class-conditional image generation and incorporates a modulation mechanism to condition the network based on both the diffusion process timestep and the class label. Different from DALLE-3 (Betker et al., 2023), we report results on more comprehensive benchmarks instead of only CLIP score, such as GenEval (Ghosh et al., 2024) and DSG (Cho et al., 2024).

**Effect of synthetic captions.** Synthetic captions lead to significant improvements on the GenEval benchmark (Ghosh et al., 2024), as shown in Table 6, highlighting the advantage of enhanced prompt-following capabilities. Notably, incorporating SSC or DSC with AltText boosts the GenEval

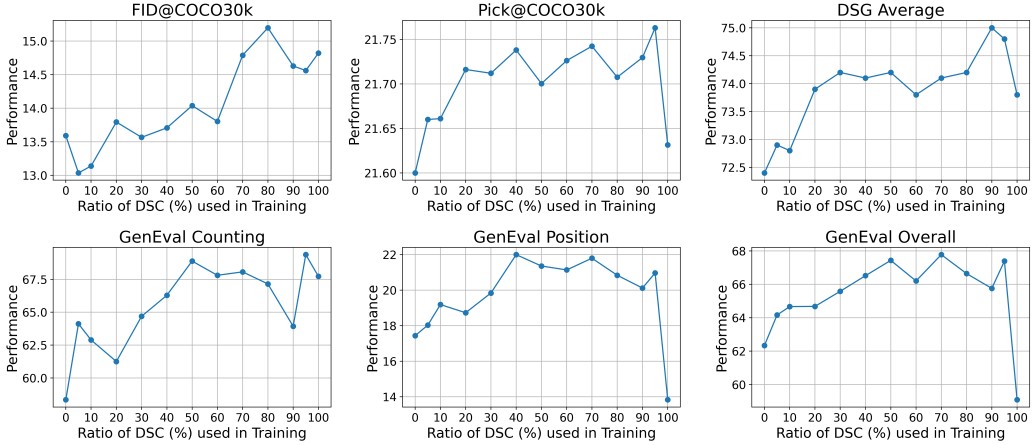

Figure 8: The intersection of synthetic captions and AltText for the diffusion model: we increase the proportion of DSC mixed with AltText during training.

average score from 58.8 to 65.5. Additionally, synthetic captions yield over a 3.5% improvement on DSG (Cho et al., 2024). However, SSC achieves better performance on the FID score (Jayasumana et al., 2024).

**Ablation study on mixing ratio of synthetic captions and AltText.** We examine the impact of varying the ratio between DSC and AltText in diffusion model training, evaluating performance across FID@COCO30k (Jayasumana et al., 2024), CLIP@COCO30k, DSG Average (Cho et al., 2024), GenEval Overall (Ghosh et al., 2024). Results are summarized in Fig. 8. The FID@COCO30k metric shows a gradual increase, suggesting that higher DSC ratios lead to improvements in generation quality. The DSG Average score exhibits improvements with a higher DSC ratio, indicating that DSC can enhance the model's ability to handle complex tasks. However, the performance on the GenEval related metric peaks at 50% DSC, after which it begins to decline, highlighting the necessity of balancing synthetic and original captions to achieve optimal results across diverse evaluation merics. Besides DSC, we further conduct experiments using SSC as well. Results are summarized in Table 6. Overall, the use of SSC alsone also achieves competitive performance, but the use of DSC and AltText together appears to be a better captioning strategy.

Table 6: The effect of synthetic captions on diffusion models.

| | GenEval (Ghosh et al., 2024) | | | | | | FID | DSG |
| | Single Obj | Two Obj | Counting | Colors | Position | Attribution | Average | COCO30k | Average |
|---|---|---|---|---|---|---|---|---|---|
| AltText | 99.1 | 80.4 | 58.3 | 78.5 | 17.4 | 40.2 | 62.3 | 13.6 | 72.4 |
| SSC | 98.5 | 80.1 | 68.9 | 80.4 | 18.6 | 50.2 | 66.1 | 13.1 | 73.1 |
| SSC + AltText | 98.2 | 84.4 | 65.1 | 77.3 | 18.6 | 50.2 | 65.5 | 13.1 | 74.2 |
| DSC | 93.9 | 60.2 | 67.7 | 79.8 | 13.8 | 39.2 | 59.1 | 14.8 | 73.8 |
| DSC + AltText | 99.2 | 84.2 | 68.9 | 79.0 | 21.4 | 52.0 | 67.4 | 14.0 | 74.2 |

## 5 DISCUSSION

In this study, we examine the role and value of image-text data in multimodal foundation models, including CLIP, multimodal LLMs, and diffusion models. Our research focuses on the intersection between synthetic image-aligned captions and the original web-crawled AltTexts. To identify the most effective captions for each foundation model, we develop a controllable and human-aligned captioning pipeline designed to minimize hallucinations and generate various types of captions as needed. Through extensive pre-training experiments, we derive the following key insights: 1) both AltTexts and synthetic captions play crucial roles—AltTexts contribute diverse information, while synthetic captions offer improved image-text alignment; 2) CLIP tends to favor short synthetic captions, whereas multimodal LLMs benefit from more descriptive captions. We also observe that the benchmarks in the pre-training and SFT stage of MLLMs may have different preferences of captions; 3) we verify the observation from DALLE-3 on text-to-image generation with more comprehensive benchmarks and show the benefits of synthetic captions.

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

# Appendices

We provide additional details for datasets, experimental settings, results, and analysis in the supplementary material.

## A  EXPERIMENTAL DETAILS

### A.1  CLIP

We summarize the training details in Table A1. For the pre-training stage, we pre-train models on up to 512 TPUs with JAX (Bradbury et al., 2018).

Table A1: Pre-training hyper-parameters and settings for the in-house CLIP.

| | |
|---|---|
| Batch size | 32768 |
| Image size | $224 \times 224$ (ViT-B/16) |
| Image pre-processing | long-side resizing with padding (i.e., `tf.image.resize_with_pad`) |
| Text tokenizer | T5 (Raffel et al., 2020), lowercase |
| Text maximum length | 77 |
| Steps | $435,000$ (i.e., $\sim 14B$ examples seen) |
| Optimizer | AdamW ($\beta_1 = 0.9, \beta_2 = 0.98$) |
| Peak learning rate (LR) | 0.0005 |
| LR schedule | cosine decays with linear warm-up (first 2k steps) |
| Weight decay | 0.2 |
| Dropout rate | 0.0 |

#### A.1.1  ADDITIONAL EXPERIMENTS

To further explore the performance gap between DSC and SSC on CLIP, we present two additional benchmarks to enhance the representativeness of CLIP's existing evaluations: 1) linear probing and 2) transferability between CLIP pre-trained with different captions and LLaVA-style MLLMs. Linear probing provides a direct measure of the quality and generalization capability of the representations learned by CLIP. Strong performance from a linear classifier on specific tasks indicates that the pre-trained model has effectively captured relevant and discriminative features, underscoring the robustness of its embeddings. Additionally, we assess CLIP's representation quality using LLaVA (Liu et al., 2023b) as a case study, where the vision encoder remains frozen during both pre-training and SFT stages. This makes LLaVA an ideal benchmark for evaluating the transferability and integrity of CLIP's learned representations.

We summarize the results on linear probing in Fig. 2: even though DSC and SSC shows lower zero-shot performance, they achieve comparable results to AltText after linear probing, indicating similar pre-trained representations. Furthermore, combining synthetic captions with AltText yields the best overall performance. Then we use these pre-trained vision encoders and insert them into LLaVA and complete the default pre-training and SFT stages in LLaVA (Liu et al., 2023b). All of our pre-trained CLIPs use ViT-B/16 as the backbone. We use Vicuna-1.3 as the LLM for LLaVA training and report recent benchmarks in Table A2: POPE (Li et al., 2023b), TextVQA (Singh et al., 2019b), GQA (Hudson & Manning, 2019), SciQA (Lu et al., 2022), LLaVA-Bench (Liu et al., 2023b), MME (Fu et al., 2024), and MM-Vet (Yu et al., 2024b). In this case, the combination of SSC and AltText achieves the highest overall performance, leading in 5 out of 9 columns. This highlights the critical role of both synthetic captions and original AltTexts in CLIP's pre-training: synthetic captions enhance image-text alignment, while AltTexts introduce valuable data diversity.

**Compatibility of rewritten-based methods and filtering-based methods.** Besides rewritten-based datasets like web-crawled VeCap-300M (Lai et al., 2024), which leverage recaptioning techniques to improve image-text alignment, it is essential to evaluate the compatibility between rewriting-based

Table A2: LLaVA as the benchmark for evaluating CLIP vision encoder pre-trained on different captions.

| Pre-trained Caption | POPE Avg. | TextVQA | GQA | SciQA | | LLaVA-Bench COCO | MME | | MM-Vet |
|---|---|---|---|---|---|---|---|---|---|
| | | | | IMG | Accuracy | | Perception | Cognition | |
| AltText | 84.4 | **50.2** | 59.1 | 64.3 | 64.8 | 76.4 | 1332.6 | 271.1 | 25.0 |
| DSC | 83.4 | 47.6 | 59.1 | 63.2 | 64.8 | 75.6 | 1307.8 | **312.9** | 23.9 |
| DSC + AltText | 84.6 | 49.8 | 59.9 | 63.6 | 65.3 | 77.2 | **1405.3** | 273.2 | **25.5** |
| SSC | 84.5 | 48.2 | 59.6 | 59.0 | 62.7 | 76.7 | 1343.6 | 248.6 | 22.4 |
| SSC + AltText | **84.7** | 49.4 | **60.1** | **65.3** | **65.8** | **77.9** | 1370.7 | 270.7 | 25.1 |

Table A3: Compatibility between rewritten-based and filtering-based methods. We use DFN-2B (Fang et al., 2023) as the example and train CLIP with ViT/B-16 on different captions.

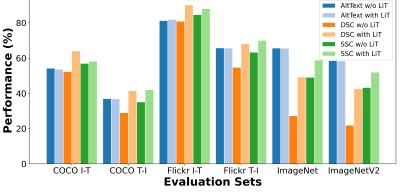

| Pre-train Caption | COCO (R@1) | | Flickr30k (R@1) | | ImageNet | ImageNetV2 |
|---|---|---|---|---|---|---|
| | I-T | T-I | I-T | T-I | | |
| AltText | 61.22 | 49.27 | 86.20 | 68.94 | **76.12** | **68.49** |
| DSC | 51.47 | 30.99 | 77.28 | 55.68 | 27.97 | 23.67 |
| SSC | 59.06 | 31.91 | 87.41 | 62.49 | 53.96 | 46.39 |
| DSC+SSC | 60.68 | 38.46 | 89.60 | 68.68 | 56.10 | 49.15 |
| AltText+DSC+SSC | **70.56** | **50.74** | **92.40** | **76.92** | 72.45 | 64.98 |

Figure A1: Effect of LiT (Zhai et al., 2022) on different captions after pre-training.

and filtering-based methods that remove mismatched image-text pairs. We consider DFN-2B (Fang et al., 2023) as a representative example, where a pre-trained CLIP model filters the dataset to retain pairs that align well with CLIP's capabilities. A key research direction is to explore whether synthetic captions can effectively replace the original CLIP-selected AltText. To investigate this, we apply our captioning pipeline to DFN-2B (Fang et al., 2023) to generate synthetic captions and pre-train CLIP models on these captions. The results are summarized in Table A3. In this CLIP-filtered dataset, neither rewritten DSC nor SSC, nor their combination, outperforms the original AltText. However, when combining all our synthetic captions with the original AltText (using uniform sampling during training), we observe significant improvements in retrieval tasks, such as **+9.34%** on COCO I-T and **+7.98%** on Flickr T-I tasks, demonstrating the enhanced image-text alignment provided by our synthetic captions. Nevertheless, incorporating synthetic captions results in a performance drop of around 4% on ImageNet, highlighting the crucial role of the diverse information contained in the original AltText for CLIP's learning.

**Effect of Locked-image text Tuning (LiT) (Zhai et al., 2022) after pre-training with different captions.** LiT (Zhai et al., 2022) trains a text model to derive meaningful representations from a pre-trained image model by freezing the image encoder and fine-tuning only the text encoder. We investigate the impact of LiT after pre-training CLIP on different captions, with the results summarized in Fig. A1. During the LiT stage, we continue to train the text encoder using AltText. Our findings reveal that LiT with AltText consistently benefits both SSC and TSC across all evaluation sets, highlighting once again the critical role of AltText in CLIP training.

## A.2 MULTIMODAL LLM

We summarize the training details in Table A4. For the pre-training stage, we pre-train models on up to 512 TPUs with JAX (Bradbury et al., 2018).

Table A4: Pre-training hyper-parameters and settings for the Multimodal LLM experiments. We use the same configuration as the 1.2B model in MM1 (McKinzie et al., 2024).

| General | |
|---|---|
| Batch size | 512 |
| Image encoder | $336 \times 336$ ViT-L/14 |
| Visual-language connector | C-Abstractor with 144 image tokens |
| Language model | 1.2B transformer decoder-only language model |
| Steps | 50000 |

For the SFT experiments, we follow the same datasets and configuration as in MM1 (McKinzie et al., 2024).

**TextCore.** For the pre-training benchmarks, TextCore is an average number of 8 benchmarks: ARC (Clark et al., 2018), PIQA (Bisk et al., 2020), LAMBADA (Paperno et al., 2016), Wino-Grande (Sakaguchi et al., 2021), HellaSWAG (Zellers et al., 2019), SciQ (Welbl et al., 2017), TriviaQA (Joshi et al., 2017), and WebQS (Berant et al., 2013).

**SFT dataset.** For the SFT benchmarks, we summarize the details in Table A5: we mainly use MME (Fu et al., 2024), SEED (Li et al., 2023a), POPE (Li et al., 2023b), LLaVA-Bench (Wild) (Liu et al., 2023b), MM-Vet (Yu et al., 2024b), TextVQA (Singh et al., 2019a), MMMU (Yue et al., 2023), MathVista (Lu et al., 2023), ScienceQA (Lu et al., 2022).

| Benchmark | Metric |
|---|---|
| MME (Fu et al., 2024) | Normalized Accuracy |
| SEED (Li et al., 2023a) | Seed-IMG |
| POPE (Li et al., 2023b) | Average of random, popular and adversarial |
| LLaVA-Bench (Wild) (Liu et al., 2023b) | GPT-assisted score |
| MM-Vet (Yu et al., 2024b) | GPT-assisted score |
| TextVQA (Singh et al., 2019a) | VQA Open Flamingo Accuracy |
| MMMU (Yue et al., 2023) | Accuracy |
| MathVista (Lu et al., 2023) | GPT-assisted score |
| ScienceQA (Lu et al., 2022) | Accuracy-IMG |

Table A5: Details of benchmarks and their metrics used in MM1.

### A.3    DIFFUSION MODEL

We summarize the training details of our self-implemented diffusion model based on Stable Diffusion 3 (Esser et al., 2024) in Table A6.

Table A6: Pre-training hyper-parameters for our diffusion model based on Stable Diffusion 3.

| | |
|---|---|
| Batch size | 4096 |
| Image size | $256 \times 256$ |
| Step | $500,000$ |
| Text condition | in house CLIP's G/14 text encoder with T5 T5 (Raffel et al., 2020) |
| Text maximum length | 77 |
| Steps | $435,000$ (i.e., $\sim 14$B examples seen) |
| Optimizer | Adafactor ($\beta_1 = 0.9, \beta_2 = 0.999$) |
| Learning rate (LR) | 0.0001 (constant with linear warm-up for 1k steps) |
| Ema decay | 0.9999 |
| Classifier free guidance | 7.5 |

## B    A DEEPER ANALYSIS OF GENERATED CAPTIONS

Fig. A2 is an overview of our two-stage fine-tuning process: we first convert a MLLM into an image captioner, then we further fine-tune it to convert it into a human-aligned captioner. Our smaller image captioning model (3B) efficiently generates large volumes of synthetic data for our experiments. Using this model, we re-captioned a dataset of 7 billion images across multiple iterations. Furthermore, our larger model, with 7 billion parameters, is designed to produce more detailed captions, surpassing the level of detail offered by our long caption format.

**AFC fine-tuning dataset.** To generate AltText Fusion Captions (AFC), we also prepare a fine-tuning dataset in this format. Specifically, given AltText and a DSC caption generated by our captioner, we

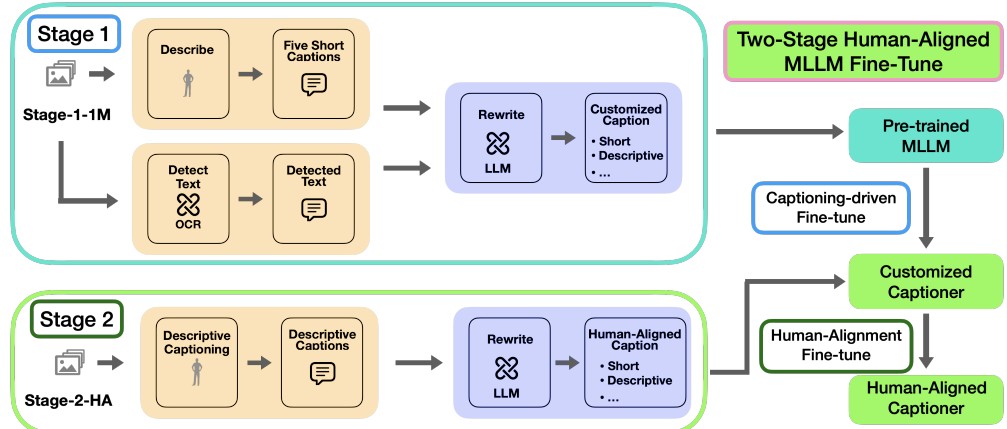

Figure A2: Overview of the controllable and human-aligned captioning pipeline. In Stage 1, we convert a pre-trained MLLM into a customized captioner that strictly follows the captioning instructions. In Stage 2, we leverage human-aligned captions to further fine-tune the captioner.

ask LLM to fuse AltText information to the DSC. By this way, we construct a 20K training dataset for our captioner.

**Less hallucinations in our DSC.** The Caption Hallucination Assessment with Image Relevance (CHAIR) metric (Rohrbach et al., 2018) is a custom-designed evaluation tool developed to identify and measure the extent of object hallucination in image captioning tasks. The metric determines the proportion of generated words that accurately correspond to objects present in the image, as verified by ground truth sentences and object segmentations. It includes two scores: one that measures the fraction of hallucinated object instances (referred to as $\text{CHAIR}_i$), and the other that calculates the fraction of sentences containing at least one hallucinated object (referred to as $\text{CHAIR}_s$):

$$\text{CHAIR}_i = \frac{|\{\text{hallucinated objects}\}|}{|\{\text{all objects mentioned}\}|}, \quad \text{CHAIR}_s = \frac{|\{\text{sentences with hallucinated object}\}|}{|\{\text{all sentences}\}|}.$$

As shown in Table A7, our model achieves a $\text{CHAIR}i$ score of 5.9 and a $\text{CHAIR}s$ score of 19.6, outperforming leading models such as LLaVA-1.5 (Liu et al., 2023b), Shikra (Chen et al., 2023), and MiniGPT-4 (Zhu et al., 2023). The lower CHAIR scores indicate that our captioner produces fewer hallucinated objects per instance and fewer sentences containing hallucinated objects. This improvement shows the effectiveness of our two-stage fine-tuning process, which strategically reduces objects hallucination by leveraging human-aligned data and strict prompt constraints. Consequently, our model offers more reliable and accurate objects recognition capabilities, making it a robust tool for generating high-quality captions across various applications.

Table A7: Hallucination detection across different MLLMs and our captioner.

| Model | $\text{CHAIR}_i(\downarrow)$ | $\text{CHAIR}_s(\downarrow)$ |
|---|---|---|
| InstructBLIP (Dai et al., 2023) | 14.5 | 30.0 |
| MiniGPT-4 (Zhu et al., 2023) | 8.2 | 24.2 |
| Shikra (Chen et al., 2023) | 7.0 | 22.0 |
| LLaVA-1.5 (Liu et al., 2023b) | 6.2 | 20.6 |
| **Our** | **5.9** | **19.6** |

**ANA: Average Number of Assertions, a metric for evaluating richness of captions.** Besides hallucination, the richness of captions are also an important index to control the generated captions. We propose ANA to quantify the richness of captions. Inspired by GenEval (Ghosh et al., 2024) for evaluating text-to-image generation models, we reverse the process of text-to-image to image-to-text. As shown in Fig. A3, we prompt an LLM to generate different assertions of a caption. After that, we can also leverage a VQA model to check if these details are aligned with the visual contents.

**CapScore: A metric for evaluating hallucinations in synthetic captions.** Although the CHAIR metric is widely used, we find it insufficient for detecting hallucinations in object attributes, especially in highly descriptive captions. To overcome this limitation, we propose a new metric to evaluate both the hallucination and richness of captions. Inspired by GenEval (Ghosh et al., 2024) for evaluating text-to-image generation models, we reverse the process of text-to-image to image-to-text and propose CapScore. CapScore measures the correctness of synthetic captions by evaluating the alignment

between generated textual assertions and the actual content of the image. As shown in Fig. A3, CapScore has two key steps: 1) use an LLM to extract structured assertions from the captions; 2) use a MLLM to serve as a VQA (Visual Question Answering) model to verify the truthfulness of these assertions. Specifically, each assertion represents a distinct factual claim made within the caption. Then the VQA model determines whether the image supports each claim by answering questions based on the assertions. CapScore is then defined as the percentage of assertions validated as correct by the VQA model. A higher CapScore indicates fewer hallucinations and greater factual accuracy in the generated captions.

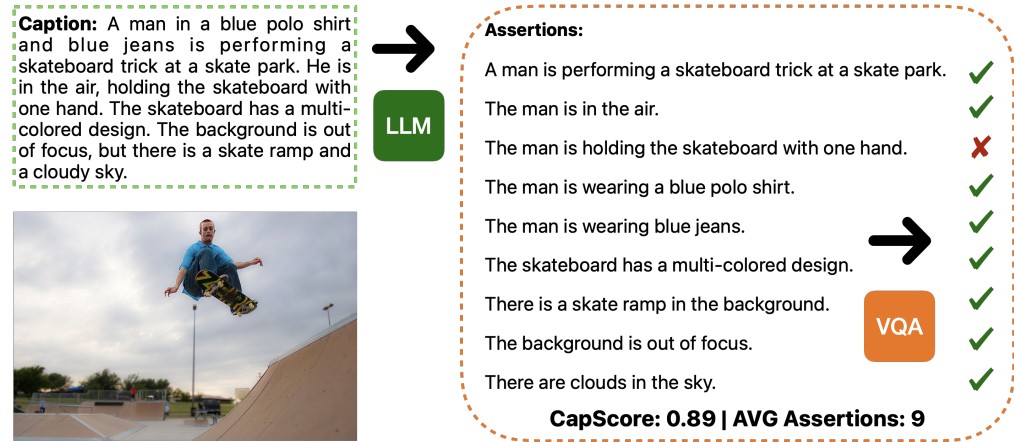

Figure A3: An overview of CapScore to evaluate the quality of captions: we use LLM to generate assertions based on the caption and then use a VQA model to check these assertions.

**Short Synthetic Caption (SSC):**
A beach with a crowd of people sunbathing and swimming in the ocean.

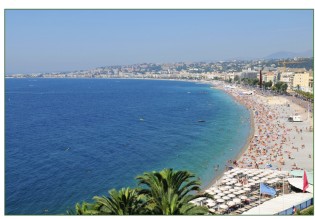

**Descriptive Synthetic Caption (DSC):**
A wide shot photograph of a beach with a crowd of people sunbathing and swimming in the ocean. The beach is surrounded by palm trees and buildings. In the background, there is a mountain and a clear blue sky.

**Dense Synthetic Caption (DSC+):**
A wide shot of a beach with numerous people on it. A group of white and blue umbrellas is on the sand on the right side. White sand and clear blue water are in front of the beach. A group of buildings are on a hill at the back of the beach, along with a hill behind it. A large crane is on top of one of the buildings on the right. It is facing to the right and slightly towards the front. Mountains are in the far background, behind the buildings. The sky is bright blue and cloudless.

**AltText Fusion Caption (AFC):**
A wide shot photograph of a beach in Nice, France on the French Riviera. The beach is crowded with people and there are many umbrellas and chairs. The ocean is a deep blue and the sky is clear. In the background, there are buildings and hills.

***AltText:*** *Bird's eye view of the French Riviera in Nice, France.*

Figure A4: Another example of controllable captions of diverse formats generated by our captioner.

As shown in Table A8, there is a notable trade-off between the richness of captions and their accuracy. As captions become longer, the Average Number of Assertions (ANA) increases, reflecting the growing richness and complexity of the generated captions. However, CapScore drops with longer captions, which suggests that while the captions provide more content, they are more prone to hallucinations—where the captioning model introduces incorrect or irrelevant information not present in the image. For instance, while DSC+ produces the highest ANA, it also demonstrates the lowest CapScore, highlighting this balance. By contrast, SSC maintains a higher CapScore with fewer assertions, demonstrating better alignment with the image content but at the cost of less detailed descriptions.

This behavior highlights the importance of balancing richness and accuracy in multimodal tasks. Models aiming for high-precision applications (e.g., zero-shot classification with CLIP) may benefit from shorter captions (e.g., SSC), whereas scenarios requiring more detailed scene descriptions (e.g., mutlimodal LLMs) may prioritize longer captions like DSC+, accepting some decrease in factual accuracy.

Table A8: CapScore and Average Number of Assertions (ANA) to evaluate the richness and accuracy of different captions.

| Caption | CapScore(↑) | ANA(↑) |
|---|---|---|
| LLaVA-1.5 (Liu et al., 2023b) | 88.76 | 7.30 |
| SSC | 91.56 | 2.49 |
| DSC | 87.30 | 8.13 |
| DSC+ | 75.74 | 12.20 |

**More detailed but noisy captions are helpful for MLLM pre-training.** We examine the impact of hallucinations in image-text data used for MLLM pre-training, focusing on the type of hallucinations detected in our captions. For our primary comparison, we select the best-performing model, LLaVA-1.5 (Liu et al., 2023b), as shown in Table A7. We first utilize LLaVA-1.5 to generate captions on the VeCap-300M dataset (Lai et al., 2024). Next, we apply our captioner to generate DSC+ for the same set of images. Our captions are more detailed but contain more hallucinations on object. We use MM1's pre-training under a small scale setting as a case study to examine whether image-caption data with fewer hallucinations or with more details can offer advantages for MLLM pre-training. We evaluate the two models after applying fine-tuning using the same data recipe.

As shown in Table A9, the MLLM pre-trained with more detailed captions performs better, even though these data contain more hallucinations. When examining task-specific results, we observe that our DSC+ pre-trained model outperforms the LLaVA captions model on 7 out of 9 benchmarks. Specifically, DSC+ improves performance on VQAv2, MMMU, MathV, and SEED, among others, indicating that the added detail from DSC+ benefits these tasks. However, despite the slight degradation in MMEP results (727.2 vs. 773.4), the overall advantage of detailed captions with minor hallucinations demonstrates that a balance between information richness and accuracy can positively impact MLLM's performance. The larger gains in multimodal understanding tasks, such as MMEC and LLaVAW, suggest that hallucination-tolerant MLLM pre-training may help with complex vision-language reasoning.

Table A9: SFT evaluation of models pre-trained with captions generated by LLaVA-1.5 and our DSC+. We use the same SFT recipe and evaluation benchmarks as in MM1 (McKinzie et al., 2024).

| Pre-Trained Data | VQAv2 | VQAT | MMMU | MathV | MMEP | MMEC | SEED | POPE | LLaVAW |
|---|---|---|---|---|---|---|---|---|---|
| LLaVA Captions | 78.0 | 65.4 | 30.0 | 27.6 | **773.4** | 200.4 | 63.5 | 83.7 | 63.6 |
| DSC+ | **79.0** | **65.4** | **32.6** | **29.4** | 727.2 | **224.3** | **66.8** | **85.1** | **71.5** |

