# OpenReview forum: "Revisit Large-Scale Image-Caption Data in Pre-training Multimodal Foundation Models"
_ICLR.cc/2025/Conference — ICLR 2025 Poster_

### Official Review · Reviewer_QQnt · 2024-10-25

**Soundness:** 3
**Presentation:** 3
**Contribution:** 2
**Rating:** 6
**Confidence:** 3

**Summary:**

This paper investigates the impact of synthetic captions on training CLIP, MLLM and a Diffusion Model. Specifically, it compares the effects of training with short versus dense captions and examines the outcomes of combining synthetic captions with original AltTexts. Additionally, the authors introduce two custom human-annotated datasets aimed at reducing hallucinations in the captioner—one tailored for short captions and the other for dense captions.

**Strengths:**

The paper addresses an interesting and important question by evaluating how captions with varying levels of visual detail impact the downstream performance of multimodal models. In addition, it approaches this evaluation more holistically than prior research by including a range of multimodal model types in the comparison.

Furthermore, the use of combined synthetic captions and original AltText represents an interesting direction that, to my knowledge, has not been extensively explored in previous studies.

**Weaknesses:**

While the paper addresses a relevant topic, several of the experiments appear similar to those in prior research, which limits the novelty of the findings. For example, as the authors note, Betker et al. [1] conducted a similar experiment evaluating the effects of long versus short synthetic captions on the downstream performance of diffusion models, reaching comparable conclusions. Similarly, Chen et al. [2] presented an experiment analogous to the MLLM analysis here, examining the impact of long versus short synthetic captions on downstream tasks. Although the models differ, the ideas and conclusions appear similar.


Additionally, while the authors justify training a custom captioner on a new dataset by claiming existing multimodal LLMs are prone to hallucinations, no experiments directly compare the effects of training with captions from, for example, LLaVA [3] or TinyLLaVA [4] (or newer and stronger models) versus the proposed captioner. Including such a comparison would enhance the reliability of the results. Moreover, the decision to create custom datasets is unclear, as similar datasets for short [3] and long captions [4] already exist. Further clarification on how the proposed captioning dataset differs from these existing resources would strengthen the paper.


[1] Betker et al. Improving Image Generation with Better Captions. 2023

[2] Chen et al. ShareGPT4V: Improving Large Multi-modal Models with Better Captions. 2024

[3] Chen et al. Microsoft coco captions: Data collection and evaluation server. 2014

[4] Onoe et al. DOCCI: Descriptions of Connected and Contrasting Images. 2024

**Questions:**

It seems like the following citations is missing? The paper explores the effect of synthetic versus AltText captions when training CLIP.

[1] Nguyen et al. Improving multimodal datasets with image captioning. 2023

---

> ### Author Response · Authors · 2024-11-21
> **Reply to Reviewer QQnt**
>
> We thank the reviewer for acknowledging the comprehensiveness of our work on the pre-training stage across a range of multimodal model types. Below are our responses to the specific questions raised:
>
> ### 1. Differences with Prior Works
> We thank the reviewer for raising this point and for highlighting relevant prior work. While we acknowledge the similarities in themes with Betker et al. [1] and Chen et al. [2], we believe our study provides several key contributions that distinguish it from these works:
>
> - **Comprehensive Investigation of Synthetic Captions:**
> Rather than focusing solely on the dichotomy of short versus long synthetic captions, our work conducts a comprehensive study aimed at first verifying the necessity and value of synthetic captions in pre-training multimodal foundation models. Additionally, we examine their interplay with original AltTexts, providing deeper insights into their complementary roles. Building on this foundation, we explore different synthetic captioning strategies and propose the novel AltText Fusion Caption (AFC) format, which balances the strengths of both short and long captions.
>
> - **Key Differences with Betker et al. [1]:**
>   While Betker et al. focused on specific benchmarks relevant to diffusion models, our study expands the scope by incorporating diverse and recent benchmarks, such as DSG and GenEval. These benchmarks provide broader and more reliable evaluations of synthetic captions. Moreover, we evaluate performance across multiple foundational models (e.g., CLIP, MLLMs, and Diffusion Models), showcasing the wider applicability of our findings.
>
> - **Key Differences with Chen et al. [2]:**
>   - Our study scales the exploration of synthetic captions to a 300M dataset, offering insights into their efficacy at a scale representative of real-world pre-training scenarios.
>   - Unlike Chen et al., who primarily analyzed the impact of captions on downstream tasks, our study delves into the pre-training stage, investigating how synthetic captions influence model initialization and subsequent performance.
>
> ### 2. Effect of Training with Captions from LLaVA
> We thank the reviewer for this excellent point. We also utilized a public model (LLaVA-13B) to caption 300M images in WIT-300M. Table A9 in the Appendix summarizes the comparison between our captions (generated by a 3B model) and those generated by LLaVA-13B.
> Our findings show that the model pre-trained on our captions outperforms the one pre-trained on LLaVA-13B captions on 7 out of 9 benchmarks. Additionally, experiments on CLIP demonstrate that our captions, despite being generated by a smaller 3B model, achieve results comparable to or better than those generated by LLaVA-13B.
> | **Caption**           | **COCO (I-T)** | **COCO (T-I)** | **Flickr (I-T)** | **Flickr (T-I)** | **ImageNet** | **ImageNetV2** |
> |------------------------|----------------|----------------|------------------|------------------|------------------|--------------------|
> | **AltText**           | 54.24          | 36.98          | 81.30           | 65.80           | 65.70           | 58.58             |
> | **LLaVA**             | 60.36          | 39.58          | 83.95           | 63.62           | 30.32           | 26.43             |
> | **Our**   | 57.00          | 35.15          | 84.67           | 63.35           | 49.1           | 43.31             |
> | **LLaVA + AltText**   | 66.81          | 48.14          | 90.17           | 75.23           | 66.76           | 59.90             |
> | **Our + AltText**   | 69.88         | 50.28          | 93.50           |77.19          | 65.16           | 58.11            |
>
> We will revise the manuscript to move this analysis to the main paper for better clarity and emphasis.
>
> ### 3. Motivation for Creating Custom Datasets
> We appreciate the reviewer’s question on this topic. While datasets like [3] (300k short captions) and [4] (15k captions) are valuable for fine-tuning, they are insufficient for large-scale pre-training of multimodal foundation models like CLIP, multimodal LLMs, and diffusion models. For example, OpenAI’s CLIP required 400M image-caption pairs for effective training.
>
> A significant contribution of our work is the development of a scalable captioning pipeline capable of generating data at the scale required for pre-training. This scalability allows us to meet the demands of pre-training multimodal models. In the revised manuscript, we will emphasize this distinction and clarify how our dataset complements and advances beyond existing resources.
>
> ---
>
> We will revise the manuscript to address these comments. We thank the reviewer again for helping us highlight the contributions of this work.

---

> > ### Comment · Reviewer_QQnt · 2024-11-22
> >
> > I thank the authors for the response and clarifications.
> >
> > Regarding the second point, I still think there is a gap between the claim in the paper that the improvements made by the custom captioner are due to reduced hallucinations, as other factors can contribute as well (e.g., caption style, difference in caption length between LLaVA13B and DSC+, etc.), please consider softening this claim.
> >
> > As my concerns are mostly addressed, I raise my score accordingly.

---

> > > ### Author Response · Authors · 2024-11-22
> > > **Reply to Reviewer QQnt**
> > >
> > > We thank the reviewer for their valuable feedback and for helping us improve our manuscript. We agree that multiple factors, such as caption style and length, could contribute to the observed improvements. We will carefully revise the manuscript to soften the claim and provide a discussion of all potential factors.

---

### Official Review · Reviewer_5ZLH · 2024-11-02

**Soundness:** 3
**Presentation:** 3
**Contribution:** 2
**Rating:** 6
**Confidence:** 3

**Summary:**

This paper investigates the interplay between alt text and synthetic captions as training data for training multimodal foundation models. It finds that overall, a mixture between alt text, short, and long synthetic captions is advantageous, with different preferences between models and tasks.

**Strengths:**

Overall the methodology appears sound and comprehensive. The array of tests provides insight into how to best combine caption data sources for multimodal training. I found particularly interesting the conclusions about the differing types of information provided by each – e.g. CLIP benefiting from the diversity of short alt texts while MLLMs may derive further benefit from longer, descriptive captions.

**Weaknesses:**

The synthetic captions are derived from a custom model (Section 3) and thus it is not clear to what extent the results would generalize to synthetic data produced by other models or those used in other works.

**Questions:**

Similarly to your AFC, have you tried rewriting or fusing alt texts to form short synthetic captions?

---

> ### Author Response · Authors · 2024-11-21
> **Reply to Reviewer 5ZLH**
>
> ## Response to Reviewer Comments
>
> We sincerely appreciate the reviewer for acknowledging the comprehensiveness of our work. Below are our responses to the specific questions raised:
>
> ### 1. Generalizability of Results Using Synthetic Data from Other Models
>
> Thank you for raising this insightful point. To address generalizability, we conducted additional experiments using the publicly available **LLaVA-13B** model to generate captions for the WIT-300M dataset (300M images). The results, as shown below, demonstrate that:
>
> - **LLaVA-generated captions** improve performance on retrieval tasks but reduce accuracy on ImageNet classification.
> - **Combining AltText with LLaVA captions** yields the best overall performance across both retrieval and classification tasks.
>
> These findings align with the trends observed when using captions generated by our **3B captioner**, supporting the robustness of our conclusions across different captioning models. The results from LLaVA-generated captions have already been discussed in the main paper (e.g., Figure 2a), and we will further highlight these findings in the revised manuscript.
>
> | **Caption**           | **COCO (I-T)** | **COCO (T-I)** | **Flickr (I-T)** | **Flickr (T-I)** | **ImageNet** | **ImageNetV2** |
> |------------------------|----------------|----------------|------------------|------------------|------------------|--------------------|
> | **AltText**           | 54.24          | 36.98          | 81.30           | 65.80           | 65.70           | 58.58             |
> | **LLaVA**             | 60.36          | 39.58          | 83.95           | 63.62           | 30.32           | 26.43             |
> | **AltText + LLaVA**   | **66.81**          | **48.14**          | **90.17**           | **75.23**           | **66.76**           | **59.90**             |
>
>
>
>
> ### 2. Short Synthetic Captions Fused with AltText
>
> Thank you for this question. Yes, we experimented with leveraging LLMs to summarize AltTexts and synthetic captions into concise short captions. However, we found it challenging to preserve both the richness of AltTexts and the semantic details of synthetic captions in a single short caption. As a result, this approach was not included in the main paper. We will add a discussion of this observation in the revised version.

---

> > ### Comment · Reviewer_5ZLH · 2024-11-24
> >
> > Thank you for your response. This helps clarify the generalizability of results beyond a specific captioning model. I also agree with QQnt's point regarding the source of improvements due to the custom captioner (claims about improved performance due to reduced hallucinations may be softened or further analyzed).
> > I will maintain my (accept) rating.

---

### Official Review · Reviewer_aQXA · 2024-11-02

**Soundness:** 3
**Presentation:** 2
**Contribution:** 2
**Rating:** 6
**Confidence:** 4

**Summary:**

The paper studies the impact of using synthetic captions in foundational vision-language models such as contrastive models, multimodal LLMs and diffusion models. The paper presents hierarchies of synthetic captions, SSC, DSC and DSC+ based on the density and length of captions and studies them across 3 broad categories of VL models. It presents results on a combination of baseline AltText and synthetic captions to understand ratios and setups that affect model performance.

**Strengths:**

1. The paper studies an important problem : understanding the impact of synthetic captions (a rapidly emerging paradigm) in conjunction with the baseline AltText captions.
2. For the models studied, such as CLIP, MM1 and SD3, the authors perform comprehensive evaluations on a wide-range of tasks. For CLIP, the authors study performance on retrieval and classification, for MM1, the authors study VQA and hallucination benchmarks and for diffusion the authors study object-centric and image fidelity metrics.

**Weaknesses:**

1. My biggest concern is that the findings are not generalizable enough.
(i) For example, findings on CLIP will not be applicable to SigLIP [1], MM1 results will not translate to Qwen-VL [2], and findings on SD3 cannot be applied on the PixArt [3] family of models. Since the paper seeks to present insights into the impact of synthetic captions, the authors should perform, for each family of models, experiments on other models which vary widely in their architecture, loss functions etc.
(ii) The synthetic captions adopted in this work is a function on top of human generated captions. Will all of the results still be valid if a different sampling of the human annotators is performed?

2. The findings are not necessary novel; for example, the fact that a mix of 50% synthetic+AltText captions for Diffusion models has been well reported in RECAP [4] and SPRIGHT [5]

[1] https://arxiv.org/abs/2303.15343
[2] https://arxiv.org/abs/2308.12966
[3] https://arxiv.org/abs/2310.00426
[4] https://arxiv.org/abs/2310.16656
[5] https://arxiv.org/abs/2404.01197

**Questions:**

1. What is the reason of using synthetic captions as a LLM re-write on top of the human captions? I notice the results in Table A7-9, and only LLM-generated captions are not far away from the approach taken in the paper. In fact, I am confident if better MLLMs such as Llava-1.6 and CogVLM are used, this gap in performance would be closer.
2.  The captioner is MM1 and the MLLM used is MM1. Is there an impact if another captioner was used?
3. zero-shot results of DSC can be significantly improved with linear probing -- what sort of linear probing is performed? If linear probing helps and is independent of the type of captions used, then do the zero-shot findings still hold?
4. Some information on the demographics of the human captioners leveraged would be helpful.
5. The reason why DSC is restricted to 78 tokens is because CLIP has the same token limit. If CLIP was not used, would both DSC and DSC+ be needed?
6. Are there any findings if Stage-1-1M and Stage-2-HA are reversed?
7. The Average Number of Assertions (ANA) pipeline is very similar to https://arxiv.org/pdf/2311.01477; are there any major differences?

---

> ### Author Response · Authors · 2024-11-21
> **Reply to Reviewer aQXA**
>
> We are grateful that the reviewer acknowledged the importance of the problem and comprehensive evaluations. Here is our response to the raised questions.
>
> ### 1. Generalizability of Results
>
> First, we respectfully argue that the concern about generalizability may not fully apply in this context given we are studying the compute-intensive pre-training. Our large-scale pre-training experiments are intensive and far from trivial. Our study focuses on a fundamental question: the role of original AltText and synthetic captions in the pre-training stages of multimodal foundation model, and aim to provide data recipe.
>
> Second, in the recent literature, we found that the data recipe is highly transferable among the same-family models. For example, the recipe of Cambrian-1 [7] is fully transferable to MM1.5 [6]. Therefore, we pick one representative model in each family and study the pre-training stage.
>
> Third, we thank the reviewer for sharing these works [1,2,3]. However, it seems these works are not open-sourced for the pre-training. We will add the discussion on more models in the Future Work in our revised manuscript.
>
> ### 2. Human Annotation
>
> Yes, we also directly use LLaVA-13B to generate captions that are independent of human annotation. We run experiments on both CLIP and multimodal LLM, where we observed the same conclusion as our captions. (See the results in Table A9 in appendix and our reply to Reviewer 5ZLH).
>
> ### 3. Findings Are Not Novel
> Thank you for sharing this point. While prior works such as RECAP [4] and SPRIGHT [5] have explored mixtures of AltText and synthetic captions, our study introduces several novel contributions.
> - We separately investigate the value of AltText and synthetic captions to better understand their individual contributions. This foundational analysis motivates us to generate diverse types of synthetic captions tailored to different use cases.
> - In contrast to [4,5], we propose and evaluate additional caption formats, such as detailed captions (DSC+) and AltText Fusion Caption (AFC), to further study the complementary nature of synthetic captions and AltText.
> - Both [4] and [5] focus exclusively on text-to-image generation models, whereas our work considers a broader range of multimodal foundation models, including CLIP and multimodal large language models (MLLMs), extending the scope of analysis beyond generation tasks.
>
> ### Question 1
>
> We appreciate the reviewer’s thoughtful feedback. Our study aims to explore the optimal synthetic captions for various multimodal foundation models, necessitating the generation of diverse caption types, such as SSC, DSC, DSC+, and AFC. To achieve this, we employ an LLM re-write mechanism to systematically prepare these captions. Regarding performance, it is important to note that our captioner is a relatively compact 3B model. A significant contribution of our work lies in the development of a scalable captioning pipeline capable of generating billion-level pre-training volumes of image-caption data. Also, human annotation can help us prepare a cleaner fine-tuning dataset.
>
> We acknowledge that employing larger and more advanced MLLMs could potentially narrow the performance gap further. However, our focus has been on balancing computational efficiency and scalability, ensuring the feasibility of generating pre-training-scale datasets. On the other hand, we want to highlight that a stronger multimodal LLM may not indicate a better captioner, as discussed in Figure 4 and Sec 3.1. We will highlight these considerations and clarify the rationale behind our design choices in the revised manuscript.
>
> ### Question 2: Another Captioner for MM1
> Thank you for the insightful question. To evaluate the impact of using a different captioner, we conducted additional experiments with a publicly available model, LLaVA-13B, generating captions for 300M images in the WIT-300M dataset. The results below demonstrate that LLaVA-generated captions improve performance in few-shot settings, which aligns with the observations reported in our main paper. Similar trends were observed when testing these captions on CLIP, reinforcing the generalizability of our conclusions across different captioning models.
>
> | Type of Captions | TextCore | 0-Shot | 4-Shot | 8-Shot |
> |------------------|----------|--------|--------|--------|
> | LLaVA captions   | 53.85    | 30.993 | 59.777 | 63.635 |
> | Alt-Text         | 53.48    | 34.804 | 55.812 | 59.723 |
>
>
>
>
>
>
>
>
>
>
> [6] MM1.5 https://arxiv.org/abs/2409.20566
> [7] Cambrian-1 https://arxiv.org/abs/2406.16860

---

> ### Author Response · Authors · 2024-11-21
> **Reply to Reviewer aQXA (Cont'd)**
>
> ### Question 3: Linear Probing
> We follow CLIP (https://arxiv.org/pdf/2103.00020) to perform linear probing: use the features before the linear projection to the embedding space and then train a logistic regression classifier. Linear probing is a way to measure the quality of embedding learning. The linear probing analysis shows the value of DSC captions, suggesting that the relatively weaker results in the zero-shot setting could be attributed to limitations or biases inherent in the benchmark rather than a deficiency in the captions themselves. We will clarify this explanation in our revised manuscript to address any potential confusion.
>
> ### Question 4: Demographics
> Thank you for the suggestion. While we acknowledge that demographic information about human captioners can be relevant in some contexts, our work primarily focuses on the interaction between AltText and synthetic captions and their impact on multimodal foundation models. The demographic background of the human captioners is not expected to influence the conclusions of this study: we also generated captions based on LLaVA-13B and the results consistently lead to similar conclusions, demonstrating that our findings are independent of the specific captioning techniques used.
>
> ### Question 5: Token Issue and DSC/DSC+
> Thank you for raising this question. While the 78-token limit for DSC aligns with CLIP's constraints, our study is **not** restricted to CLIP but more foundation models. The DSC/DSC+ formats were intentionally designed to explore the optimal captioning strategies for various multimodal foundation models, each with unique capabilities and token-handling limits. Models like multimodal LLMs (e.g., MM1) can process much longer captions. By proposing DSC+ and AFC, we aim to examine how caption format and detail impact performance in systems that are not constrained by shorter token limits. This allows us to investigate the trade-offs between conciseness and descriptiveness across models with differing architectures and token limits.
>
> ### Question 6: Reverse 2 Stages
> Stage-2-HA comprises high-quality human-annotated captions. Consistent with prior works, we fine-tune the captioner progressively, starting with lower-quality captions (Stage-1-1M) and moving to higher-quality captions (Stage-2-HA) to ensure effective training. Reversing these stages may not align with this progressive refinement strategy. On the other hand, the captioner itself is not the primary contribution of our work. Our focus is on studying the interaction between AltText and synthetic captions and identifying optimal synthetic caption formats for various multimodal foundation models.
>
> ### Question 7: ANA
> Thank you for pointing this out. The key difference lies in our approach to quantifying caption richness. Instead of relying on sentence length, which does not necessarily correlate with the level of detail, we introduce ANA as a more precise metric. Assertions, as discrete factual statements, better capture the richness of a caption and provide a clearer basis for quantifying hallucinations. This distinction allows our method to assess captions with consideration of both richness and hallucination.
>
> We will revise the manuscript to address these comments. We thank the reviewer again for helping us revise/polish the manuscript and highlight the contributions of this work.

---

> > ### Comment · Reviewer_aQXA · 2024-11-22
> > **Response to Authors**
> >
> > Thank you for your detailed rebuttal!
> >
> > My thoughts on the response to the weaknesses :
> >
> > 1. I still find that the findings are limited. For example, the impact of re-captioning on CLIP has also been studied before in https://arxiv.org/abs/2305.20088.
> > 2. The issue with generalization remains. While I understand that all of these experiments are very computationally expensive, I suggest the authors to have a smaller experimental setting, with multiple models of varying degrees. It will help the paper be more useful to the wider community. Btw, I think SigLIP (https://huggingface.co/docs/transformers/main/en/model_doc/siglip) , Qwen-VL (https://github.com/QwenLM/Qwen-VL) and Pixart-Alpha (https://pixart-alpha.github.io/) are all open-sourced for training.
> >
> > My thoughts on the response to the questions :
> >
> > 1. My primary question was : why were the captions a re-write on top of the human captions ? Could the authors please clarify that?
> > 2. Thanks for the experiments on LLaVA!
> > 3. Thanks for the explanation on linear probing! Non-linear probing could be an interesting addition to the paper too. Something along these lines : https://arxiv.org/pdf/2404.08636
> > 4. I think some discussion on the demographics of the users that wrote the captions would be an useful addition to the paper.
> > 5. I understand the DSC+ is "denser" than DSC, and the fact that DSC is limited to 78 tokens makes it seem like its created only for CLIP-like models. A finer distinction of this density of captions could be explored to make the claims stronger.
> > 6. Thanks for the update. It still might be interesting to explore the reversal of order.
> > 7. Thank you for the clarity.
> >
> >
> > Overall, I think the paper would be a good addition to the community. I still think the findings are not necessarily very novel, but the authors have done a comprehensive job in the experimental settings they have considered. Thus, I increase my score.

---

> > > ### Author Response · Authors · 2024-11-23
> > > **Thank you for the great suggestions!**
> > >
> > > We thank the reviewer for the great feedback.
> > >
> > > Weakness 1: Thank for sharing this work. We agree that re-captioning on CLIP has been studied in the previous work and cited this work (LaCLIP) in Line 147-148. Here is the main difference between these works and our work: LaCLIP used a LLM to rewrite the AltText, which may be limited as rewriting based on raw and noisy captions cannot introduce new image-relevant information. Therefore, though it keeps the information from AltText, the visual information may be missed if the raw captions are noisy. We propose AFC to use an image captioner and then fuse AltText into the final caption. Second, different from existing work, we systematically studied the role of synthetic captions and AltTexts in pre-training, and their relationship, which is important for future research.
> > >
> > > Weakness 2 : Thank for the great suggestion. We will list these works and discuss them in the Future Work.
> > >
> > > Question 1: Thank you for the question. Human annotation is costly, so we only asked annotators to provide detailed image descriptions (annotate out one caption per image) to train the captioner for DSC. To explore the effects of various caption formats (e.g., SSC, AFC), we reformatted these human-annotated captions using an LLM to create fine-tuning datasets for all formats. We appreciate this point and will clarify it in the revised manuscript.
> > >
> > > Question 4: Thanks for the this great point. We will work with the labeling team and gather/discuss demographic information upon legal approval of our institution.
> > >
> > > Question 5: Thanks a lot for this great point. We intentionally designed DSC with a 78-token limit to optimize for CLIP-like models
> > > as they are very popular and are used to build a wide range of more advanced multimodal foundation models, e.g., diffusion models use CLIP's text encoder.  We agree with the reviewer that we need a finer distinction for DSC as serving for CLIP-like models and the foundation models building upon CLIP's text encoder.
> > >
> > > Question 3 & 6: Thank you for these excellent suggestions. We will address them in the Future Work and Limitations sections. Regarding reversed training the captioner, we agree with the reviewer. However, retraining the captioner, generating captions for large-scale over 300M images, and pre-training foundation models on them is infeasible within the discussion period. Thus, we will include this as Future Work.
> > >
> > > We sincerely thank the reviewer again for their excellent feedback and for helping us improve our manuscript.

---

### Official Review · Reviewer_EoMe · 2024-11-04

**Soundness:** 3
**Presentation:** 3
**Contribution:** 3
**Rating:** 6
**Confidence:** 3

**Summary:**

This work studies the roles of (1) short description caption (2) dense caption, and (3) AltText in training multi-modal models.

**Strengths:**

- The paper is well-motivated and grounded in a thorough analysis of various captioned datasets used for training multimodal models. This analysis offers valuable insights that effectively inform the proposed solution.
- The paper makes an important conclusion that AltText is valuable in training MLLMs, providing guidelines for future research.
- The experimental studies are comprehensive, encompassing three different multimodal models: CLIP-style models, ViT-LLM models, and diffusion models.

**Weaknesses:**

- While the performance of models trained on different data ratios is presented, there is no guidance on how to determine the optimal ratio.
- There is no discussion regarding the release of the trained captioning models or the datasets, which could limit the impact of this work.

**Questions:**

The human-annotated Stage-1-1M and Stage-2-HA datasets are valuable, and could be easily hosted on HuggingFace. Will these datasets be made available for future research?

---

> ### Author Response · Authors · 2024-11-21
> **Reply to Reviewer EoMe**
>
> We sincerely thank the reviewer for acknowledging the _thoroughness and comprehensiveness_ of our experiments and analysis, which span three diverse multimodal models. As noted, all our experiments are conducted during the _pre-training stage_, which requires substantial computational resources and is far from trivial. We deeply appreciate the reviewer’s recognition of the effort and significance of this work.
>
> Below are our responses to the two key points raised by the reviewer:
>
> ### 1. **Optimal Ratio**
>
> Thank you for the constructive feedback. We agree that determining the optimal ratio of synthetic captions to original AltText is a challenging yet essential problem.  Our study primarily focuses on investigating the role and format of synthetic captions in the pre-training stage, addressing two core questions:
> 1. What is the role of synthetic captions and AltText in pre-training? What is their relationship?
> 2. What is the most effective format for generating synthetic captions?
>
> We verified that incorporating synthetic captions during pre-training consistently improves the performance of multimodal foundation models across diverse benchmarks. This finding highlights their value as an augmentation to original AltText, enriching the diversity and semantic depth of the training data.  Through our experiments, we explored various captioning styles (e.g., SSC, DSC, DSC+, AFC) and identified how these formats impact downstream performance, contributing to a better understanding of optimal data design.
>
> Regarding the optimal ratio, our results reveal that:
> - The ratio is _task-dependent_ and _context-sensitive_, as no universal ratio works for all downstream tasks.
> - Extreme ratios (e.g., entirely synthetic or entirely original captions) tend to underperform.
>
> For example, as shown in Figure 6, a ratio between 50-90 yields consistently strong performance for CLIP. We will include this high-level guidance in the manuscript to provide actionable insights for practitioners.  By presenting these trends, we aim to offer a framework for informed decision-making, while acknowledging that further ratio tuning may be required for specific tasks or domains.
>
> ### 2. **Model/Data Release**
>
> Thank you for your comment. Our work primarily focuses on analyzing the role and relationships between AltText and synthetic captions, providing data recipes for pre-training multimodal foundation models. While the development or release of captioning models or datasets is not the primary focus of our study, we recognize the value of sharing such resources. We hope to release the model / datasets and are working with our legal team, but it will upon legal approval of our institution. We hope that our data recipe contributes to advancing the research community's understanding and development of multimodal foundation models.

---

### Meta-Review · Area_Chair_3s8T · 2024-12-11

**Metareview:**

The paper explores the contributions of synthetic and alt-text captions for pretraining MLLMs. There are some concerns about novelty (prior work by Betker) and generalizability (with respect to using different captioning models), with the main contribution lying in the benchmarking with these underexplored types of data (e..g synthetic data). These concerns are addressed sufficiently and all reviewers lean weakly (borderline) positive.

**Additional Comments On Reviewer Discussion:**

Multiple reviewers comment that their concerns have been addressed by the author responses.

---

### Decision · Program_Chairs · 2025-01-22

Accept (Poster)